# Retrospective Definition of *Clostridioides difficile* PCR Ribotypes on the Basis of Whole Genome Polymorphisms: A Proof of Principle Study

**DOI:** 10.3390/diagnostics10121078

**Published:** 2020-12-12

**Authors:** Manisha Goyal, Lysiane Hauben, Hannes Pouseele, Magali Jaillard, Katrien De Bruyne, Alex van Belkum, Richard Goering

**Affiliations:** 1BioMérieux, Open Innovation and Partnerships, 3 Route du Port Michaud, 38390 La Balme Les Grottes, France; manisha.goyal@biomerieux.com; 2BioMérieux, Applied Maths NV, 9830 Sint-Martens-Latem, Belgium; lysiane.hauben@biomerieux.com (L.H.); katrien.debruyne@biomerieux.com (K.D.B.); 3BioMérieux, Industry, 69290 Craponne, France; hannes.pouseele@biomerieux.com; 4BioMérieux, 69280 Marcy l’Etoile, France; magali.dancette@biomerieux.com; 5Department of Medical Microbiology and Immunology, Creighton University School of Medicine, 2500 California Plaza, Omaha, NE 68178, USA; richardgoering@creighton.edu

**Keywords:** ribotyping, *C. difficile*, unitigs, markers, WGS, GWAS, CDI, in silico PCR

## Abstract

*Clostridioides difficile* is a cause of health care-associated infections. The epidemiological study of *C. difficile* infection (CDI) traditionally involves PCR ribotyping. However, ribotyping will be increasingly replaced by whole genome sequencing (WGS). This implies that WGS types need correlation with classical ribotypes (RTs) in order to perform retrospective clinical studies. Here, we selected genomes of hyper-virulent *C. difficile* strains of RT001, RT017, RT027, RT078, and RT106 to try and identify new discriminatory markers using in silico ribotyping PCR and De Bruijn graph-based Genome Wide Association Studies (DBGWAS). First, in silico ribotyping PCR was performed using reference primer sequences and 30 *C. difficile* genomes of the five different RTs identified above. Second, discriminatory genomic markers were sought with DBGWAS using a set of 160 independent *C. difficile* genomes (14 ribotypes). RT-specific genetic polymorphisms were annotated and validated for their specificity and sensitivity against a larger dataset of 2425 *C. difficile* genomes covering 132 different RTs. In silico PCR ribotyping was unsuccessful due to non-specific or missing theoretical RT PCR fragments. More successfully, DBGWAS discovered a total of 47 new markers (13 in RT017, 12 in RT078, 9 in RT106, 7 in RT027, and 6 in RT001) with minimum q-values of 0 to 7.40 × 10^−5^, indicating excellent marker selectivity. The specificity and sensitivity of individual markers ranged between 0.92 and 1.0 but increased to 1 by combining two markers, hence providing undisputed RT identification based on a single genome sequence. Markers were scattered throughout the *C. difficile* genome in intra- and intergenic regions. We propose here a set of new genomic polymorphisms that efficiently identify five hyper-virulent RTs utilizing WGS data only. Further studies need to show whether this initial proof-of-principle observation can be extended to all 600 existing RTs.

## 1. Introduction

*Clostridioides difficile* (*C. difficile*), formerly known as *Clostridium difficile*, is an anaerobic, spore-forming Gram-positive bacterial species that can survive in harsh environments. It can withstand high temperatures, exposure to ultraviolet light, toxic chemicals, and exposure to antibiotics. Colonization by *C. difficile* is asymptomatic. The development of disease is mostly driven by host factors and disruption of the gut microbiome by frequent consumption of antibiotics [1,2,3]. Toxigenic strains of *C. difficile* can be a lethal cause of *C. difficile* infection (CDI), which is commonly associated with post antibiotic diarrhea [4,5]. *C. difficile* is present in the environment and can be transmitted to patients or healthcare workers through contact with contaminated surfaces. Inter-human spread mainly occurs through the fecal–oral route. *C. difficile* spores are intrinsically resistant to antibiotics and remain viable during antibiotic treatment. Clindamycin, cephalosporins, and fluoroquinolones are considered as major antibiotics associated with CDI [6]. Food or water contamination, gastric acid-suppression, and asymptomatic carriage in the community are the potential risk factors of community acquired CDI [7]. One-third of the total CDI burden occurring in the USA in 2011 was community-associated [8]. CDI caused half a million hospital-acquired infections and 29,000 deaths in 2012 in the United States [8] and approximately 40,000 cases of CDI in Europe [9]. Only a limited number of studies reported emerging CDI in Asia [10]. The increasing incidence of CDI and rapid evolution of antibiotic resistance in *C. difficile* has become a global threat to public health [11,12,13].

CDI diagnosis allows early pathogen isolation and treatment of infection, thereby reducing the potential of CDI transmission. Various diagnostic procedures for CDI are available, including toxigenic culture, cell cytotoxic neutralization assay, glutamate dehydrogenase assay, the detection of toxins by enzyme immunoassays, nucleic acid amplification-based molecular tests, etc. [14,15,16,17,18,19]. Still, epidemiological *C. difficile* strain typing is necessary to identify outbreaks within a hospital or the wider community and facilitates understanding of the dissemination of infections. Ribotyping is a classical technique for *C. difficile* typing initially based on hybridization patterns of conserved ribosomal RNA probe sequences [20,21]. Ribotype (RT) analysis has also been extremely important in the long-term surveillance of CDI [22]. While traditional typing methods such as restriction endonuclease analysis (REA) and pulsed-field gel electrophoresis (PFGE) were widely used in the past, PCR-based ribotyping is the current method of choice for *C. difficile* typing [23,24]. PCR ribotyping is dependent on the amplification of the intergenic spacer region (ISR) between 16S and 23S rRNA genes [25,26,27,28]. Since most bacterial species encode multiple ribosomal alleles in their genomes, multiple fragments of different lengths are amplified when different species but also different strains are considered [25,26]. There are still considerable constraints on PCR ribotyping including elevated costs, a higher probability of false-positive results, and a lack of interlaboratory portability [10,29,30].

Bacterial whole genome sequencing (WGS) has the potential to provide more detailed epidemiological information than classical PCR ribotyping [23,31]. To further explore a WGS-based approach to *C. difficile* typing, backward compatibility with PCR ribotyping is essential [32]. Previous studies reported the association of RT001, RT017, and RT027 with lethal CDI and considered those isolates as hyper-virulent [33,34]. A survey conducted in the North East of England concluded that RT001, RT027, and RT106 were among the most prevalent and dangerous clones [35]. In the United States and Europe, RT001, RT014, RT020, RT027, and RT078 have been identified as predominant [36,37]. RT017 is a globally emerging toxigenic RT and can be found on almost every continent [38,39]. Thus, here, we tested both in silico PCR ribotyping and the De Bruijn graph-based Genome Wide Association Study (DBGWAS) [40] for their capacity to perform retrospective PCR ribotyping for *C. difficile* RT001, RT017, RT027, RT078, and RT106. These strains were chosen as a test set representing global, long-term circulating and clinically relevant epidemic strains. The primary study goal was to develop a proof-of-principle procedure for sequence-based *C. difficile* strain typing with retrospective compatibility to established PCR RTs.

## 2. Materials and Methods

### 2.1. In Silico PCR-Based Ribotyping

We performed in silico PCR using canonical ribotyping PCR primers. Based on the reference sequences 16S-USA and 23S-USA (Table 1), in silico PCR was performed using the subsequence search tool in BioNumerics v7.6 software (Applied Maths NV, Sint Martens-Latem, Belgium). Besides 7 genomes obtained from Creighton University, 23 *C. difficile* genomes of five selected RTs were downloaded from NCBI to verify in silico amplification of the ISR region (Appendix A). 

### 2.2. DBGWAS-Mediated Discovery New RT-Specific Markers

A total number of 160 *C. difficile* genome assemblies (training set) of 14 different RTs including hyper-virulent RT001, RT017, RT027, RT078, and RT106 were used for the discovery of unique RT genomic markers (Table 2). This small training set allowed for the development of discriminatory markers to characterize the five major RTs among the 14 different RTs. Genomes were collected from the National Center for Biotechnology and Information (NCBI; www.ncbi.nlm.nih.gov), Creighton University, and the Enterobase databases (https://enterobase.warwick.ac.uk). Metadata of these genomes are summarized in Appendix A. 

To identify associations between variant genetic loci and PCR RT, a hypothesis-free DBGWAS method was used. DBGWAS defines genetic variants linked to phenotypic traits via single nucleotide polymorphism (SNP), insertions, deletions, and consequences of recombination [41,42]. We used an open source tool (https://gitlab.com/leoisl/dbgwas) [40]. The tool is able to cover variants in coding as well as non-coding regions of bacterial genomes. DBGWAS was performed keeping the tool parameters in the default setting for different *C. difficile* ribotypes (RT001, RT017, RT027, RT078, and RT106) and their RT-specific genetic variants observed in the training set. Each *C. difficile* RT was considered independently in this study. DBGWAS identified short signature sequences called (overlapping) k-mers, yielding a compact summary of all variations across a set of genomes [40]. Overlapping k-mers are called unitigs and were selected on the basis of their specific and unique presence in a particular RT. Q-values define test sensitivity and specificity and are Benjamini–Hochberg-transformed *p*-values for controlling the false-positive results in case of multiple testing [40,43]. R scripting was used to deal with large matrices defining the presence (1) or absence (0) of extracted unitigs in the training set of *C. difficile* genomes.

### 2.3. Validation of Markers

Validation of novel unitig markers was performed by means of BLAST searches against the test set of genomes (Table 3). A wide range of 2425 genomes covering 132 different *C. difficile* RTs was downloaded [44] and processed using a Linux shell script. These sequences represented PCR ribotyped strains from different countries and clinical and environmental specimens for which phylogenetic analyses were already performed by Frentrup et al. [44] (Appendix A). A database of this test set was created to perform local command line BLAST searches against the set of significant unitigs identified above. The specificity of all the unitigs was tested using strict parametric filters of 100% coverage and identity.

### 2.4. Statistically Reliable Ribotype Prediction

To evaluate the potential typing significance of the unitigs as compared to the classical ribotyping of *C. difficile* strains, sensitivity and specificity (selectivity) were computed for all the unitigs [45]. The efficiency of GWAS can be measured by assessing the false discovery rate (FDR) [46]. To increase the potential typing significance of our new method, combination statistics were performed. Sensitivity and specificity were also computed for certain combinations of two or more selected unitig sequences. The parameters defined were, next to the FDR, TP (true-positives, correctly predicting positive values, e.g., number of true RT017 predicted as RT017), FP (false-positives, missed negative values, e.g., number of non-RT017 genomes still predicted as RT017), FN (false-negatives, missed positive values), and TN (true-negatives, correctly rejected values). 

### 2.5. Functional Annotation of Unitigs

Selected unitigs were annotated using BLASTn alignment. Well-characterized *C. difficile* genomes were used as a reference to locate these new markers. Specific annotation for each marker was filtered out using minimum E-value, 100% identity, 100% coverage, and 0 gap score.

## 3. Results and Discussion

### 3.1. In Silico PCR

The visualization of amplified DNA sequences from the intergenic region between 16S and 23S ribosomal genes is the current Gold Standard for *C. difficile* typing [25,47]. In our study, in silico PCR for 30 randomly selected, well-characterized *C. difficile* genome sequences was essentially unsuccessful (Figure 1 and Appendix A). Genome sequences included generated insufficient numbers of differently sized fragments. The fragment sizes that were calculated were verified with the online tool available at http://insilico.ehu.es/PCR [10]. On the other hand, more recently sequenced *C. difficile* genomes were showing only one or even none of the expected amplified fragments (Appendix A). There is a substantial possibility that the PCR ribotyping fragments observed upon laboratory experimentation for these strains may not derive from ISR variants but rather from random amplification. Thus, in silico PCR failed to generate reliable RTs which prompted us to explore the feasibility of DBGWAS-based typing. Of note, we presume here that NGS-based methods are very likely to be more reliable than any of the many other molecular typing methods.

### 3.2. New Genotyping Markers

A total number of 47 RT-specific unitigs (13 for RT017, 12 for RT078, 9 for RT106, 7 for RT027, and 6 for RT001) were identified. The unitigs shared an average length of 56 base pairs (Table 4). DBGWAS generated compacted De Bruijn graphs (cDBG) containing the specific unitigs as nodes defining a genotypic association between a particular RT and the *C. difficile* genomes included (Figure 2). Unitigs that were specific for a particular RT were color-coded according to their association to the RT (red for positive association, blue for negative association) and minimum q-values were provided by subgraphs. Q-values for selected unitigs ranged from 0 to 7.40 × 10^−5^. Significantly associated unitigs were extracted as FASTA formatted sequences (Appendix A).

### 3.3. Validation of Markers

Unitigs showing 100% identity in all genomes belonging to a single RT in the validation set demonstrated the efficiency of these unique patterns to carry out in silico ribotyping. Although the individual unitig-based characterization of *C. difficile* strains was not absolute, it allowed RT determination with approximate sensitivity and specificity of between 0.90 and 1.0 (Figure 3). FDR for all the unitigs for RT017 was the lowest (0.06) followed by RT027 (0.08), RT078 (0.23), RT001 (0.30), and RT106 (0.46) (Figure 3).

Some of the unitigs were shared by closely related RTs. Unitigs for RT001 were able to identify the genetically closely related RT087, RT241, and RT012, which altogether form a clonal complex (CC) 141 [44]. One of the markers identified for RT017 showed no false-positives or false negatives. Other markers for RT017 initially generated a small number of false-positives, but 100% true-positives in the validation dataset. Markers for RT078 identified 78 out of 79 isolates of RT126 and all of the RT413 strains from the test dataset, likely due to the close genetic relatedness of these RTs (CC 1) [44,48,49]. Unitig sequences for RT106 were also able to identify *C. difficile* RT500 along with RT106 from the test set. Phylogenetic grouping of *C. difficile* genomes [44] showed that *C. difficile* core genome multi locus sequence typing (cgMLST) of RT106 and RT500 (CC 22) generated completely indistinguishable groupings. Considering closely related strains as true-positives based on their respective RTs, the FDRs for each unitig subset were found to be smaller, underscoring the biological consistency of the results. Adding genomes of RT413, RT126, and RT500 to the training set resulted in a decreased FDR rate. The continuously increasing number of publicly available *C. difficile* genome sequences will provide substantial opportunities for improvement of our new characterization technique. 

### 3.4. Marker Combination Study

For the ribotypes RT027, RT078, RT106, and RT001, every possible combination of RT-specific unitigs was created and tested for statistical significance. A combination of two unitigs was shown to increase sensitivity and specificity up to 1 and to reduce the FDR to 0.05 (Figure 4A–D). Each combination was defined on the basis of logical operators “AND/OR”. The AND operator symbolizes that both the markers in a combination need to be present with 100% identity, whereas the OR operator means that any one of the two markers in a combination need to be present at one time, again with 100% sequence identity. There is no combination required in the case of RT017. Conclusively, as clearly exemplified in Figure 4A–D, in certain cases, the combination of markers improves RT testing by suppression of the false discovery rate. Marker’s SEQ ID numbers and their sequences are given in Appendix A.

### 3.5. Functional Annotation of Markers

Functional characterization of the regions from which our unitigs originated demonstrated that 34% of the unitigs were localized in intergenic regions (five for RT027, four for RT001, three for of RT017 and RT106 each, and one for RT078 (Figure 5, Figure 6, Figure 7, Figure 8 and Figure 9). Six percent of all markers were left unannotated in RT001, RT027, and RT078 (one marker for each) (Table 4). Only RT001 was identified with a unitig marker residing within the rRNA-23S ribosomal gene showing at least some correspondence with ribotyping (Figure 5). This marker did not show sufficient diagnostic power and was thus not selected in the final set of markers. All other markers were observed to be scattered throughout the *C. difficile* genome. In RT078, one of these markers was identified in a mobile genetic element (Figure 8). Mostly, genes and intergenic regions, apart from the conserved ribosomal ISR, were observed to play a potential role in the unitig-mediated *C. difficile* typing. 

## 4. Conclusions

Strain typing has a proven value in monitoring the persistence and spread of bacterial pathogens in human populations. For *C. difficile*, PCR ribotyping is the current first choice but may be challenged now that genome sequencing is an option. No single-step test or algorithm is available so far for correlating *C. difficile* RTs with WGS data. This implies that there may be an issue with the correlation between WGS-based epidemiological analysis and PCR ribotyping for *C. difficile*. Here, we show that DBGWAS identified unique genomic markers that would suit that specific purpose. A combination of two unitigs led to 100% sensitive and specific discrimination between five important RTs. We believe that this approach is highly promising, providing a clear opportunity to define backward compatibility between classical RTs and WGS data.

## Figures and Tables

**Figure 1 diagnostics-10-01078-f001:**
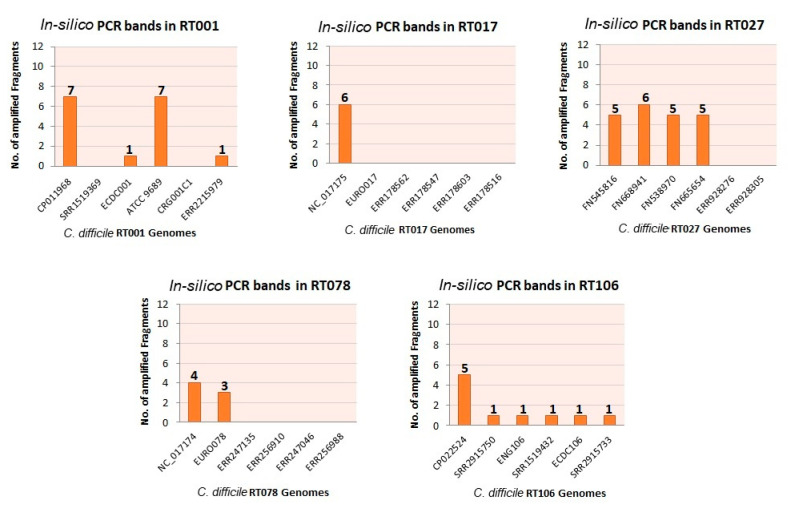
In silico ribotyping of different *C. difficile* genome sequences using the ISR 16S and 23S USA primer pair. The five panels represent the results obtained for examples of five different ribotypes. Bar graphs show the number of theoretical PCR bands (vertical axis, number of bands labeled on each bar) in the ribosomal region of respective genome sequences (horizontal axis), whereas the genomes without any fragments depict the complete absence of primer binding sites in those genomes. Note that the expected outcome would be an identical number of fragments for each of the strains belonging to a single ribotype. We indicated this number as the first marker of reproducibility; it has to be stated that besides this variation of numbers of fragments, the size of the fragment was also determined as a variable as well.

**Figure 2 diagnostics-10-01078-f002:**
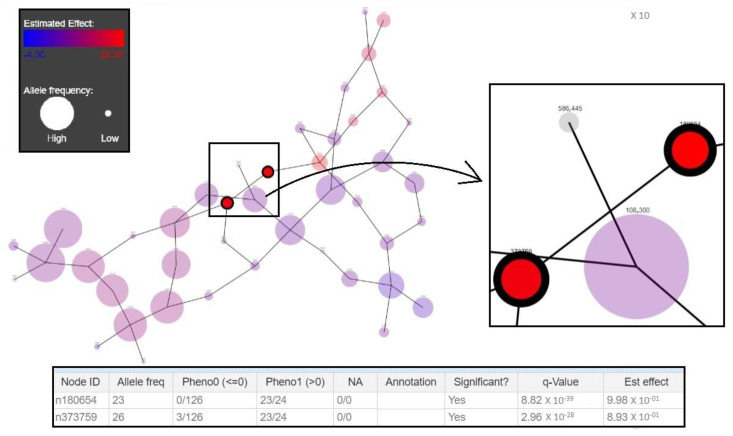
Compacted De Bruijn graphs (cDBG) generated by De Bruijn graph-based Genome Wide Association Studies (DBGWAS) for RT001 genome sequences. The figure illustrates the significance of the nodes (representing the selective sequences called unitigs), which are denoted by their estimated effect ranging from high (28.304; red) to low (4.00; blue). Allele frequency is represented by the size of the node. The table explains that from the two selected significant nodes in terms of their association with ribotype, the node on the top right (n180654) is specific to RT001 (called Pheno 1 in the table) and completely absent in the other ribotypes in the training set (Pheno 0). Additionally, the q-value linked to the first node is very significantly below 0.05 and hence, the estimated effect is high (represented by the red color of the node).

**Figure 3 diagnostics-10-01078-f003:**
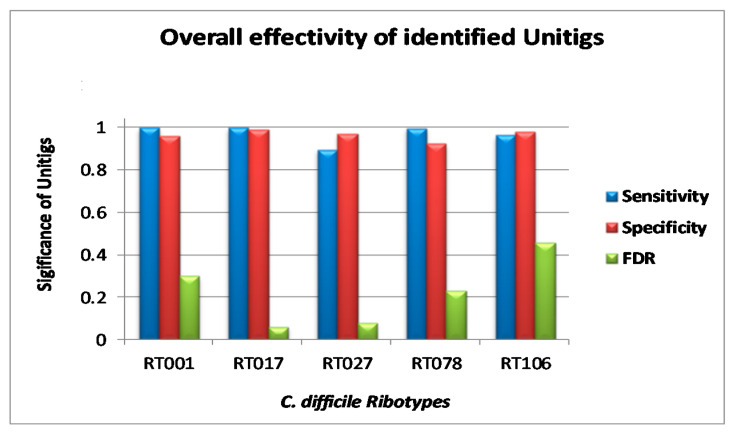
Statistical comparison of genome typing efficiency of discovered unique patterns for selected *C. difficile* ribotypes in terms of specificity, sensitivity, and false discovery rate (FDR).

**Figure 4 diagnostics-10-01078-f004:**
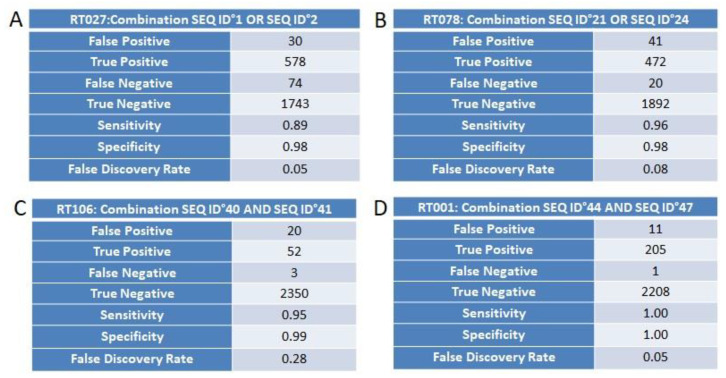
(**A**–**D**) Statistical reliability in terms of sensitivity, specificity, and false discovery rate (FDR) for the combination of two selected markers using OR operator for the identification of *C. difficile* RT027 (Panel **A**) and RT078 (Panel **B**). Panels **C** and **D** display similar analyses but then using the AND operator for identification of RT106 and RT001, respectively.

**Figure 5 diagnostics-10-01078-f005:**
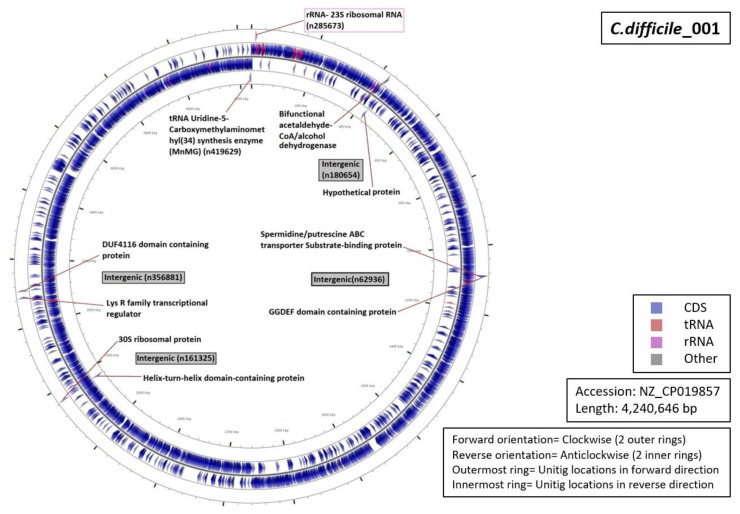
Functional annotation and location of DBGWAS markers on the reference genome of *C. difficile* RT001. Both central rings represent the genome annotation (reverse inside, forward outside), while the outer and inner rings represent the signature sequences (unitigs) (reverse inside, forward outside).

**Figure 6 diagnostics-10-01078-f006:**
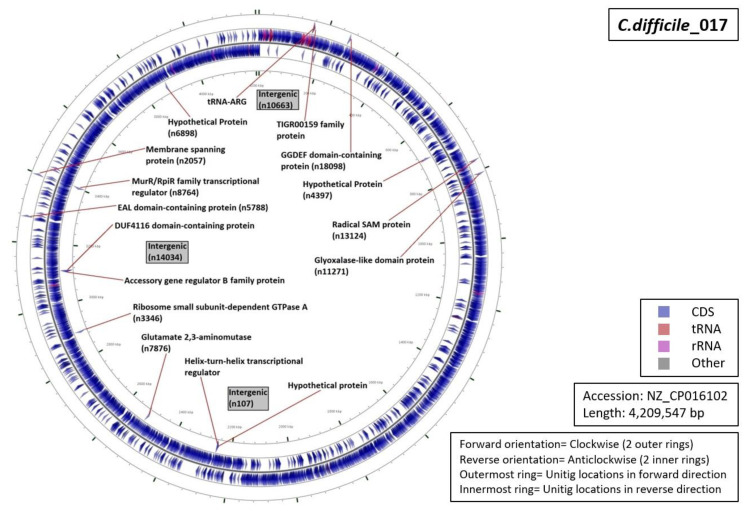
Functional annotation and location of DBGWAS markers on the reference genome of *C. difficile* RT017.

**Figure 7 diagnostics-10-01078-f007:**
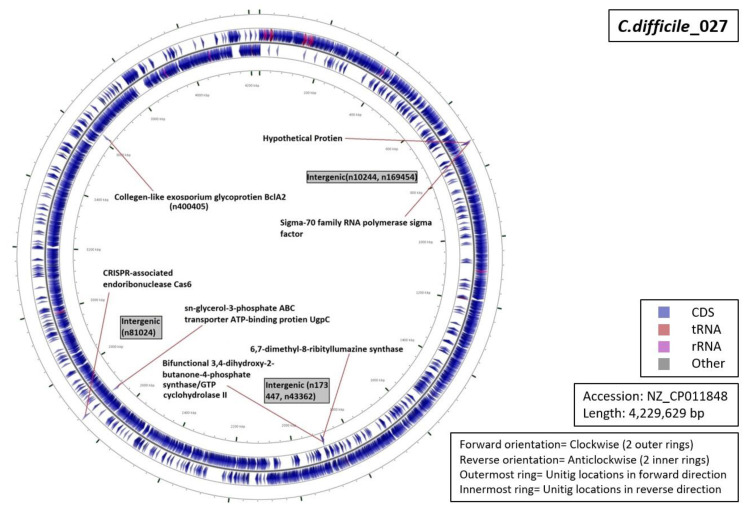
Functional annotation and location of DBGWAS markers on the reference genome of *C. difficile* RT027.

**Figure 8 diagnostics-10-01078-f008:**
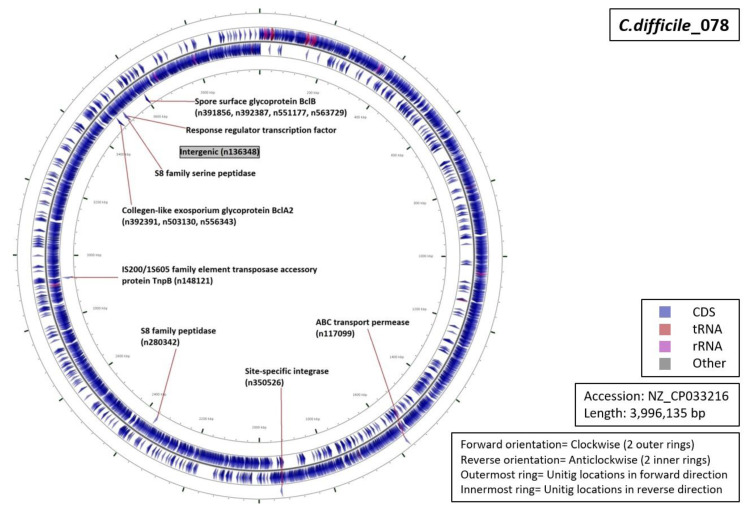
Functional annotation and location of DBGWAS markers on the reference genome of *C. difficile* RT078.

**Figure 9 diagnostics-10-01078-f009:**
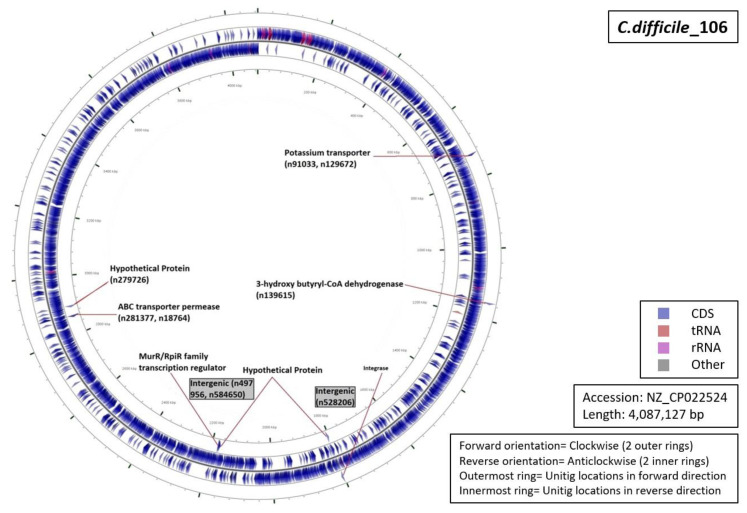
Functional annotation and location of DBGWAS markers on the reference genome of *C. difficile* RT106.

**Table 1 diagnostics-10-01078-t001:** Primer pair used for in silico PCR-based ribotyping of *Clostridioides difficile*.

Primer	Gene Target	GenBank Accession No.	Sequence (5’–3’)	Tm (°C)	Reference
16S-USA (Forward)	16S rRNA gene	FN545816	(12293)GTGCGGCTGGATCACCTCCT (12312)	71.0	Xiao et al., 2012 (46)
23S-USA (Reverse)	23S rRNA gene	FN545816	(12621)CCCTGCACCCTTAATAACTTGACC (12598)	67.1	

**Table 2 diagnostics-10-01078-t002:** *C. difficile* ribotypes included in the training dataset along with the number of genomes and their source of availability.

*C. difficile* Ribotype	Number of Genomes	Source
RT001	24	Enterobase, NCBI, Creighton University
RT002	2	NCBI, Creighton University
RT003	19	NCBI, Creighton University
RT005	19	NCBI, Creighton University
RT010	3	NCBI, Creighton University
RT014	11	NCBI, Creighton University
RT015	2	NCBI, Creighton University
RT017	15	NCBI, Creighton University
RT023	3	NCBI, Creighton University
RT027	15	NCBI, Creighton University
RT046	4	NCBI, Creighton University
RT078	15	NCBI, Creighton University
RT106	22	Enterobase, NCBI, Creighton University
RT126	6	NCBI, Creighton University
**TOTAL**	**160**	

**Table 3 diagnostics-10-01078-t003:** *C. difficile* ribotypes downloaded from the Enterobase database as a test dataset and the number of genomes included in each ribotype.

Ribotype	Count	Ribotype	Count	Ribotype	Count	Ribotype	Count
RT001	206	RT046	3	RT127	1	RT375	1
RT002	53	RT049	9	RT129	1	RT404	15
RT003	11	RT050	4	RT137	1	RT413	13
RT005	14	RT051	1	RT138	1	RT446	2
RT006	1	RT053	5	RT149	1	RT449	2
RT009	2	RT054	2	RT150	1	RT451	1
RT010	7	RT056	5	RT153	1	RT453	1
RT011	3	RT058	1	RT156	1	RT454	1
RT012	45	RT060	1	RT157	1	RT456	1
RT013	1	RT062	2	RT158	1	RT470	1
RT014	113	RT063	1	RT176	13	RT500	21
RT015	36	RT066	3	RT193	1	RT534	1
RT017	272	RT067	1	RT194	1	RT547	1
RT018	55	RT069	1	RT212	1	RT559	1
RT019	1	RT070	4	RT220	4	RT563	1
RT020	44	RT072	1	RT225	1	RT569	1
RT022	1	RT073	2	RT226	1	RT581	1
RT023	16	RT075	1	RT236	3	RT585	1
RT024	1	RT076	2	RT238	1	RT586	1
RT026	6	RT077	1	RT239	2	RT591	1
RT027	652	RT078	492	RT241	5	RT598	8
RT029	3	RT081	2	RT244	9	RT614	1
RT031	2	RT083	1	RT251	1	RT620	2
RT032	1	RT084	2	RT262	1	RT629	1
RT033	5	RT087	8	RT284	1	RT651	1
RT035	2	RT090	1	RT289	1	RT666	1
RT036	1	RT094	1	RT290	1	RT668	1
RT037	1	RT102	1	RT305	1	RT678	1
RT039	5	RT103	2	RT316	1	RT708	1
RT042	2	RT106	55	RT321	1	RT719	1
RT043	2	RT117	2	RT328	2	RT720	1
RT044	2	RT125	1	RT336	1	RT721	1
RT045	2	RT126	79	RT356	8	RT722	1

**Table 4 diagnostics-10-01078-t004:** Number of unique markers identified for each *C. difficile* ribotype, their average length, and annotation.

Ribotype	Number of Markers	Average Length (Base Pairs)	Annotation (Number of Unitigs)
RT001	06	59	Intergenic (4) tRNA uridine-5-carboxymethylaminomethyl synthesis enzyme MnmG (1) rRNA-23S ribosomal RNA (1 excluded from the list) Unknown (1)
RT017	13	69	Intergenic (3) Membrane spanning protein (1) Ribosome small subunit-dependent GTPase A (1) Hypothetical protein (1) EAL domain-containing protein (2) Glutamate 2,3-aminomutase (1) MurR/RpiR family transcriptional regulator (1) GGDEF domain-containing protein (1) Glyoxalase-like domain protein (1) Radical SAM protein (1)
RT027	07	53	Collagen-like exosporium glycoprotein BclA2 (1) Intergenic (5) Unknown (1)
RT078	12	42	IS200/IS605 family element transposase accessory protein TnpB (1) Spore surface glycoprotein BclB (4) Collagen-like exosporium glycoprotein (BclA2) (3) Unknown (partial with ABC transporter permease) (1) Intergenic (1) Site-specific integrase (1) S8 family peptidase (1)
RT106	09	55	Intergenic (3) ABC transporter permease (2) Hypothetical Protein (1) 3-Hydroxybutyryl-CoA dehydrogenase (1) Potassium transporter (2)

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
