# Peer review of "Retrospective Definition of Clostridioides difficile PCR Ribotypes on the Basis of Whole Genome Polymorphisms: A Proof of Principle Study"

_diagnostics, 2020, doi:10.3390/diagnostics10121078_

Round 1
Reviewer 1 Report
Whole genome sequencing is replacing slowly, but surely, all others classical molecular typing methods. However, when a specific typing method is wildly used for outbreak investigation and surveillance of a specific microorganism, it is necessary to make the link between “classical types” that were identified by the classic method and results of WGS.
- difficile ribotyping is a very good example, as toxigenic epidemic ribotypes have been wildly described in the scientific and public health communities. Therefore, the present study is of utmost importance. However, when I read the sentence at line 154 to 156, it revealed something that I was suspecting. If the conclusion of the authors is that there is a substantial possibility that PCR ribotyping fragments observed in the lab are from random amplification, this means that the method is not reliable and we should not rely on studies using this method! Moreover, the authors consider lab ribotyping as the gold standard in their study; if this method is not reliable, how can they be sure that investigated genomes have the correct ribotype in their metadata? Authors should comment on this.
A phylogenetic analysis (core SNP calling, wgMLST) should have been performed first on these over 2000 genomes in order to assess the phylogenetic basis of ribotypes, with possible curation (?).
Line 45. As Cdiff is part of the microbiota, “is asymptomatic” would be more appropriate.
Line 70. The reference of Bidet et al. 1999 should be added, as it is the first description of the primers that are currently used for ribotyping.
Line 71. The principal of ribotyping of Cdiff should be explained in more details in order to understand the in silico ribotyping performed in this study. E.g. As the rRNA operon on the bacterial chromosome is in multi copy, the PCR amplifies as many DNA fragment of different sizes as there are different alleles of this operon. Differences in alleles were shown to be due to the presence of a repeat region of 9 pb (Indra 2010 JMM).
Line 74 to 87. According to the authors, seven ribotypes (RT001, 014, 017, 020, 027, 078, 106) are predominantly reported. Why did they not include RT014 and RT020 in their study?
Line 83. A reference should be given for DBGWAS.
Paragraph 3.1. It is difficult to judge if the in silico PCRs failed or not. Indeed, expected laboratory size of DNA fragments are not given for each investigated ribotype and comparison with in silico results is not possible. The presented results show only on that genomes from the same ribotype give different number of presumptive bands with in silico PCRs. Moreover, it took me several time to understand this. Supp Table 1 is difficult to understand; the note is incomprehensible. I would suggest to have one column per expected band, to report the fragments that were defined in each band’s cell, and to add the expected pattern for each ribotype :
Thus, Figure 1, or a table (?) should report the number of in silico bands identical to laboratory bands and the number of discrepancies. Authors should consider to rewrite this paragraph 3.1.
English of Figure 2 legend should be improved. What do the authors means by “phenotype”? Is it the lab PCR ribotype? If yes, a ribotype is a genotype, not a phenotype! Authors should clarify.
Figure 2 is of poor quality and the text is unreadable and; e.g. the number 180654 cannot be seen.
Figure 3. Values can be only between 0 and 1. The Y axis should be modified consequently.
Tables 5a to d should grouped into one table.
References should be revised; e.g title and journal name of ref 43 is missing.
English should be revised.

Author Response
COMMENTS REVIEWER 1
Whole genome sequencing is replacing slowly, but surely, all others classical molecular typing methods. However, when a specific typing method is wildly used for outbreak investigation and surveillance of a specific microorganism, it is necessary to make the link between “classical types” that were identified by the classic method and results of WGS.
C.difficile ribotyping is a very good example, as toxigenic epidemic ribotypes have been wildly described in the scientific and public health communities. Therefore, the present study is of utmost importance. However, when I read the sentence at line 154 to 156, it revealed something that I was suspecting. If the conclusion of the authors is that there is a substantial possibility that PCR ribotyping fragments observed in the lab are from random amplification, this means that the method is not reliable and we should not rely on studies using this method! Based on extensive prior experience we can only agree with the reviewer: all PCR mediated methods carry a risk of being a-specific and hence difficult to reproduce, especially when multiple centers are involved. This is one of the secondary reasons why we undertook the present study. Still, we do feel that retrospective consideration of genome sequences versus ribotypes may be useful even if it would only serve to define pathogenic clones such as RT027 among genome sequences. This may have clinically relevant information value.
Moreover, the authors consider lab ribotyping as the gold standard in their study; if this method is not reliable, how can they be sure that investigated genomes have the correct ribotype in their metadata? Authors should comment on this. Unfortunately ribotyping simply IS the current Gold Standard in the Cdif typing community, very little we can do about that now ….. Although it has its drawbacks it did help define clonal dissemination of certain types (eg RT027 as mentioned above). We have now included more explicit statements on the superiority of NGS over RT!!
A phylogenetic analysis (core SNP calling, wgMLST) should have been performed first on these over 2000 genomes in order to assess the phylogenetic basis of ribotypes, with possible curation (?). This was not the goal of our study. Our target was much more simple and straightforward. In addition, the sequences were obtained from a very well curated collection of strains and the sequences were studied in phylogenetic detail by Frentrup et al whom first described this collection (reference is in the literature list of our paper).
Line 45. As Cdiff is part of the microbiota, “is asymptomatic” would be more appropriate. Adapted as suggested.
Line 70. The reference of Bidet et al. 1999 should be added, as it is the first description of the primers that are currently used for ribotyping. Adapted as suggested.
Line 71. The principal of ribotyping of Cdiff should be explained in more details in order to understand the in silico ribotyping performed in this study. E.g. As the rRNA operon on the bacterial chromosome is in multi copy, the PCR amplifies as many DNA fragment of different sizes as there are different alleles of this operon. Differences in alleles were shown to be due to the presence of a repeat region of 9 pb (Indra 2010 JMM). Adapted as suggested. We simply copy-pasted the sentence suggested by the reviewer in the text, thanks for your help J. The Indra paper was already cited.
Line 74 to 87. According to the authors, seven ribotypes (RT001, 014, 017, 020, 027, 078, 106) are predominantly reported. Why did they not include RT014 and RT020 in their study? We selected those clones that showed the best overall global dissemination. Question with studies kike ours is always how many clones one should study to make a point and we felt that with five we would have proof of principle at least (explicitly stated in the title ….)
Line 83. A reference should be given for DBGWAS. Adapted as suggested.
Paragraph 3.1. It is difficult to judge if the in silico PCRs failed or not. Indeed, expected laboratory size of DNA fragments are not given for each investigated ribotype and comparison with in silico results is not possible. The presented results show only on that genomes from the same ribotype give different number of presumptive bands with in silico PCRs. Moreover, it took me several time to understand this. Supp Table 1 is difficult to understand; the note is incomprehensible. We have adapted the legend to an extend where we hope that the illustration is now also good enough to keep included as was. Hence we ignored the next sentence in the reviewer report. Let us know if we should elaborate further. I would suggest to have one column per expected band, to report the fragments that were defined in each band’s cell, and to add the expected pattern for each ribotype :
Thus, Figure 1, or a table (?) should report the number of in silico bands identical to laboratory bands and the number of discrepancies. Authors should consider to rewrite this paragraph 3.1. We rewrote extensively and hope that this is now more clear. See also reasoning in reply to previous question by the reviewer.
English of Figure 2 legend should be improved. What do the authors means by “phenotype”? Is it the lab PCR ribotype? If yes, a ribotype is a genotype, not a phenotype! Authors should clarify. Again, we edited significantly and hope that this is now OK.
Figure 2 is of poor quality and the text is unreadable and; e.g. the number 180654 cannot be seen. We did not notice this problem in the PDF we got. Anyway, we have now more graphically described the position of the nodes (“top left corner”) which prevents any mistake toward interpretation
Figure 3. Values can be only between 0 and 1. The Y axis should be modified consequently. Adapted as suggested.
Tables 5a to d should grouped into one table. Adapted as suggested.
References should be revised; e.g title and journal name of ref 43 is missing. We have added the title. Will the journal do a final reference check? We ask this since in the published papers in Diagnostics we saw that there is a link to the original papers embedded in the manuscript.
English should be revised. Adapted as suggested; we have put in the effort but further optimization might be possible since the largest proportion of authors are non-native English speaking.
Reviewer 2 Report
The author order is different in the submitted manuscript (Goyal is first author) as compared to the order listed above (van Belkum is first author). This needs to be resolved prior to publication. Additionally the formatting is inconsistent (text size, etc.) throughout the file.
In this paper, the authors have proposed two methods using computational approaches for differentiating C. difficile ribotypes using whole genome sequencing data that next will need to be tested experimentally. The work is relevant as C. difficile infections are highly problematic in the hospital environment and this reviewer agrees sequencing is impacting diagnostic options. The writing and figures in the paper are generally clear, however, the stated goal to correlate WGS data with ribotypes is incongruent with the second to the last sentence of the abstract in which the authors state they “propose a set of new genomic polymorphisms that efficiently identify five hyper-virulent RTs.” This could be reported as an additional goal or outcome.
The authors report upon testing in silico PCR and DBGWAS for interpreting ribotypes from sequence data. One straight-forward and simple way to report the WGS sequence variations observed for the different ribotypes would be to list the base change as compared to a reference genome. A table with this information could be reported in the paper or as a supplement. The authors state that the in silico PCR method did not work. In Figure 1, it would be helpful to report if the bands differ in base pairs and the number of base pairs in the fingerprint rather than simple number of bands. Understandably the bins will not be as specific as sequencing. They report the DBGWAS method is more promising. Showing a comparison of the DBGWAS method for two ribotypes in Figure 2 would be helpful to demonstrate the differences in the connection patterns for different RTs.
Specific comments:
Page 1, line 21: Add space after period before “First”
Page 1, line 37: Add period at end of sentence.
Page 2, line 63: Bender et al. recently published an example of a nucleic acid amplification based molecular test for C. difficile (https://doi.org/10.3390/microorganisms8040561)
Page 2, line 71: Intergenic spacer region is commonly abbreviated IGS rather than ISR.
Page 2, line 80: Capitalize States.
Page 5, line 137: Correctly is misspelled.
Table 4: Instead of reporting the average length of simulated amplicons, consider reporting the discrete band sizes.
Author Response
COMMENTS REVIEWER 2
The author order is different in the submitted manuscript (Goyal is first author) as compared to the order listed above (van Belkum is first author). This needs to be resolved prior to publication. Additionally the formatting is inconsistent (text size, etc.) throughout the file. Goyal is and will remain fist author, differences to that are due to the fact the Van Belkum did the submission and will be communicating author. We will keep a very close eye on this!! Text inconsistency is most likely due to formatting errors by the journal.
In this paper, the authors have proposed two methods using computational approaches for differentiating C. difficile ribotypes using whole genome sequencing data that next will need to be tested experimentally. The work is relevant as C. difficile infections are highly problematic in the hospital environment and this reviewer agrees sequencing is impacting diagnostic options. The writing and figures in the paper are generally clear, however, the stated goal to correlate WGS data with ribotypes is incongruent with the second to the last sentence of the abstract in which the authors state they “propose a set of new genomic polymorphisms that efficiently identify five hyper-virulent RTs.” This could be reported as an additional goal or outcome. Really do not understand this comment …..
The authors report upon testing in silico PCR and DBGWAS for interpreting ribotypes from sequence data. One straight-forward and simple way to report the WGS sequence variations observed for the different ribotypes would be to list the base change as compared to a reference genome. A table with this information could be reported in the paper or as a supplement. This might turn out to be a huge illustration and essentially all of this information can be already retrieved from the current supplementary files.
The authors state that the in silico PCR method did not work. In Figure 1, it would be helpful to report if the bands differ in base pairs and the number of base pairs in the fingerprint rather than simple number of bands. Understandably the bins will not be as specific as sequencing. They report the DBGWAS method is more promising. This comment is similar to that of Reviewer 1. We edited significantly and hope the current text is more clear.
Showing a comparison of the DBGWAS method for two ribotypes in Figure 2 would be helpful to demonstrate the differences in the connection patterns for different RTs. We feel multiple comparisons would unnecessarily complicate the figure so we ignored this comment. If deemed mandatory by the editor then we could try to propose another figure. Still, the current illustration is similar to ones published before in some of our papers. We feel this similarity might help interpretation.
Specific comments:
Page 1, line 21: Add space after period before “First”. Adapted as suggested.
Page 1, line 37: Add period at end of sentence. Adapted as suggested.
Page 2, line 63: Bender et al. recently published an example of a nucleic acid amplification based molecular test for C. difficile (https://doi.org/10.3390/microorganisms8040561) We already have five references to diagnostics included and feel that that would be enough. We also prefer the first initial key papers over more recent references/
Page 2, line 71: Intergenic spacer region is commonly abbreviated IGS rather than ISR. We do not agree and we stuck with ISR.
Page 2, line 80: Capitalize States. Adapted as suggested.
Page 5, line 137: Correctly is misspelled. Adapted as suggested.
Table 4: Instead of reporting the average length of simulated amplicons, consider reporting the discrete band sizes. We did not change since we do not see the difference between how we did it and this suggestion by the reviewer.